# A Breast Cancer Polygenic Risk Score Is Feasible for Risk Stratification in the Norwegian Population

**DOI:** 10.3390/cancers15164124

**Published:** 2023-08-16

**Authors:** Bayram Cevdet Akdeniz, Morten Mattingsdal, Mev Dominguez-Valentin, Oleksandr Frei, Alexey Shadrin, Mikk Puustusmaa, Regina Saar, Siim Sõber, Pål Møller, Ole A. Andreassen, Peeter Padrik, Eivind Hovig

**Affiliations:** 1Center for Bioinformatics, Department of Informatics, University of Oslo, 0313 Oslo, Norwayehovig@ifi.uio.no (E.H.); 2Norwegian Centre for Mental Disorders Research (NORMENT), Division of Mental Health and Addiction, Oslo University Hospital, University of Oslo, 4956 Oslo, Norwayole.andreassen@medisin.uio.no (O.A.A.); 3Department of Medical Research, Vestre Viken Hospital Trust, Bærum Hospital, 1346 Gjettum, Norway; 4Department of Tumor Biology, Institute for Cancer Research, Oslo University Hospital, 0424 Oslo, Norwaymoller.pal@gmail.com (P.M.); 5OÜ Antegenes, 50603 Tartu, Estonia; mikk.puustusmaa@antegenes.com (M.P.); regina.saar@antegenes.com (R.S.);

**Keywords:** breast cancer, polygenic risk score, age-dependent risk, Cox regression, iCare

## Abstract

**Simple Summary:**

Various genomic variants that are statistically associated with breast cancer (BC) have been discovered and robustly replicated as a result of different genome-wide association studies. Such findings have led to the development of a different risk classification with the Polygenic Risk Score (PRS). In this paper, we have calculated the PRS of the Norwegian samples using various PRS models, compared their performances, and then evaluated the PRS-based lifetime risk of developing BC. The best performing PRS model includes 3820 SNPs (AUC = 0.625 and OR = 1.567), and the other studied models also provide closer performances. The results show that the PRS can be a useful instrument for lifetime risk stratification of developing BC in the Norwegian population, and can thus be utilized in the BC screening program.

**Abstract:**

Background: Statistical associations of numerous single nucleotide polymorphisms with breast cancer (BC) have been identified in genome-wide association studies (GWAS). Recent evidence suggests that a Polygenic Risk Score (PRS) can be a useful risk stratification instrument for a BC screening strategy, and a PRS test has been developed for clinical use. The performance of the PRS is yet unknown in the Norwegian population. Aim: To evaluate the performance of PRS models for BC in a Norwegian dataset. Methods: We investigated a sample of 1053 BC cases and 7094 controls from different regions of Norway. PRS values were calculated using four PRS models, and their performance was evaluated by the area under the curve (AUC) and the odds ratio (OR). The effect of the PRS on the age of onset of BC was determined by a Cox regression model, and the lifetime absolute risk of developing BC was calculated using the iCare tool. Results: The best performing PRS model included 3820 SNPs, which yielded an AUC = 0.625 and an OR = 1.567 per one standard deviation increase. The PRS values of the samples correlate with an increased risk of BC, with a hazard ratio of 1.494 per one standard deviation increase (95% confidence interval of 1.406–1.588). The individuals in the highest decile of the PRS have at least twice the risk of developing BC compared to the individuals with a median PRS. The results in this study with Norwegian samples are coherent with the findings in the study conducted using Estonian and UK Biobank samples. Conclusion: The previously validated PRS models have a similar observed accuracy in the Norwegian data as in the UK and Estonian populations. A PRS provides a meaningful association with the age of onset of BC and lifetime risk. Therefore, as suggested in Estonia, a PRS may also be integrated into the screening strategy for BC in Norway.

## 1. Introduction

Breast cancer (BC) is one of the most common human malignancies, accounting for 25.4% of all cancers in women [1]. It is currently the leading cause of cancer deaths in women worldwide. Every year, 2 million new diagnoses and more than 600,000 deaths are reported [2]. In Europe, BC caused nearly 85,000 deaths and accounted for 16% of all female cancer deaths in 2019 [3].

Around 30% of the total BC risk has been shown to be hereditary in the Nordic countries [4]. A known genetic predisposition to BC is attributable to germline pathogenic variants in single genes—i.e., monogenic risk (monogenic pathogenic variants—MPVs) and the cumulative impact of disease risk-related single nucleotide polymorphisms (SNPs) in many genes—i.e., polygenic risk (polygenic risk score—PRS). Several of the strongest genetic risk factors for BC development are known [5,6,7]. An additional component of genetic susceptibility is a family history without known MPV and PRS data. The individual BC risk is approximately two times higher in cases of BC in first-degree relatives [8].

Most hereditary cancer pathogenic variants confer susceptibility to develop cancers in multiple organs [9]. Up to ~25% of hereditary BC can be explained by the existence of highly penetrant risk variants in the tumor suppressor genes BReast CAncer gene 1 (*BRCA1*) and BReast CAncer gene 2 (*BRCA2*) [10]. Notably, carriers of pathogenic *BRCA1* or *BRCA2* variants have an increased risk of developing BC (an average lifetime risk of 35–85%). On the other hand, these pathogenic variants are rarely observed in the population. Moreover, a recent gene-level association study presents the effects of many other genes in addition to *BRCA1* and *BRCA2* [11]. Therefore, additional genetic evaluations may be beneficial to determine the patient risk of developing BC in real-world settings. Assessments of genetic predisposition can be used to estimate individual risk levels, with the potential of applying precision, or stratified, screening and prevention management.

Several genome-wide association studies (GWAS) have been conducted to identify SNPs associated with BC. Despite the individually small effects of these common variants, it has been observed that combining the effects of these variants using a PRS approach may have a clinical utility [12]. By calculating the PRSs of the samples using summary statistics obtained from GWAS studies and applying the PRS values to perform BC stratifications, promising results have been obtained with different PRS models [5,13,14,15]. Furthermore, a PRS model associated with modified risk for carriers of moderate-penetrance genes has also been presented [16]. The clinical use of PRS in BC is an emerging topic, but before it can move from clinical studies to everyday practice, more real-world evidence across different populations is required.

A comparative risk analysis of available PRS models conducted while introducing a new PRS model with 2803 SNPs for BC has been presented, namely the AnteBC (PRS 2803) (AnteBC (PRS 2803) is developed by Antegenes and is registered as a CE-marked medical device (in vitro diagnostics) in the EUDAMED database (UDI-DI: 04745010362019), the Estonian Medical Devices Database (EMDDB: 14726), and the UK MHRA Registry (GMDN: 59918)) [17]. The prediction accuracy of the best-performing model (PRS 2803) was evaluated on independent data and developed into a clinically usable test. The time- to-event of the onset of BC was modelled dependent on the PRS values using a Cox regression survival approach [18], and a significant relation between the onset of BC and PRS values was observed. Based on individual lifetime risk estimates obtained using the PRS and hazard ratio estimates [17] with the iCare tool (an R package to model a time-dependent absolute risk model for developing BC specifically [19]), the PRS was demonstrated to be a useful stratification approach to obtain a lifetime risk of developing BC.

PRS has been proposed as a promising risk stratification tool to personalize BC screening programs, with the aim of maximizing early detection while optimizing screening costs and minimizing adverse side effects, including false positive findings [20,21]. In particular, the authors of [17] proposed a protocol using the PRS to determine the starting time and frequency of conventional screening approaches such as mammography and magnetic resonance imaging (MRI). To date, PRS analyses for risk stratification for the Norwegian population have been undertaken on a limited number of traits, including ADHD [22], schizophrenia [23], and obesity [24]. Despite its effectiveness on those traits, a PRS analysis of BC on Norwegian samples has, until now, not been presented. 

Here, we conducted a comparative PRS analysis using the available PRS models for BC on Norwegian samples. We calculated the PRS, hazard ratio, and lifetime risk estimation of developing BC. We obtained a similar performance on the prediction of BC using the PRS, as obtained in earlier studies [17]. Results also show that women with a higher PRS have a higher risk of developing BC at an earlier age, such as 35 years of age. Therefore, for women with a higher PRS, it may be beneficial to start conventional screening prior to the general recommendation (50 years) [25]. The results of our study parallel the findings in the Estonian and UK populations [17], and therefore the individual risk-based screening strategy proposed for the Estonian population may also be applicable to the Norwegian population. 

## 2. Methods and Materials 

### 2.1. Data

We used genome data obtained from a geographically scattered sample set from Norway [26]. Genomic DNA was isolated from peripheral blood samples using the DNeasy Blood & Tissue Kit (Qiagen, Germantown, MD, USA), according to the manufacturer’s protocol, and genotyped using the Illumina OmniExpress 24 v 1.1 chip at deCODE genetics (Reykjavik, Iceland). The genotyped dataset with N = 15,769 individuals and 713,014 SNPs was stored and processed at the “Services for sensitive data” (TSD) platform at the University of Oslo (Oslo, Norway). The genotyped data were filtered by removing autosomal SNPs with a missing rate > 2%, followed by the removal of SNPs with a minor allele frequency (MAF) < 2%. Finally, samples with more than 2% missing data were excluded. This resulted in 583,183 autosomal SNPs typed in 14,429 individuals remaining (for more details about the dataset used, see the Appendix A). 

For the phenotype data, 16,029 Norwegian individuals were included and 9201 had a diagnosis of cancer. The most common cancer types included skin, BC, ovary, and colon. Of these, 3223 had pathogenic germline variants in clinically actionable predisposition genes. These samples were aggregated in the 1980–1995 period. Clinical, genetic, and demographic information was available for the included cases and controls.

Some cases were present in families, and we also have family history information. Approximately 2000 samples have been whole genome sequenced to use as a local reference panel. These samples belonged to 845 families with a family history of cancer and 498 sporadic cases. For whole genome sequencing (WGS) data we used the DRAGEN-GATK pipeline [27], and the resultant WGS dataset with 1368 individuals was used as a Norwegian reference panel named Norgene. More details regarding the genome and phenome data have been presented in Appendix A.

### 2.2. Imputation

The imputation was conducted using three alternative pipelines with different reference panels. The main aim of applying multiple imputation pipelines was to evaluate the effects that imputation panels may have on the result of risk stratification of BC. We performed imputations using (1) the Norwegian reference panel, (2) the MoBa imputation pipeline [28], which uses the Haplotype Reference Consortium (HRC) as a reference panel, and (3) the Antegenes imputation pipeline [17], which uses 1000 Genome phase 3 data as the reference panel. The details of these imputation pipelines are given in Appendix A.

### 2.3. PRS Model

We chose the PRS models available in the literature for BC and which were previously tested in [17]. Those PRS are subsequently referred to by SNP count for clarity, and are listed as 77 SNPs [13], 313 SNPs [5], 3820 SNPs [5], and 2803 SNPs [17].

The PRSs were calculated using PRSICE2 software [29] for the imputed Norwegian data as the target genomic data input, and the PRS models defined above were used as the base summary statistics input. In particular, the PRS of an individual j was calculated as
$$PRS_{j}=\sum_{i=1}^{M}a_{ij}\beta_{i}\;$$
where $\beta_{i}$ is the effect size of the SNP and is obtained from the external studies mentioned above that propose the corresponding PRS models, and $a_{ij}$ corresponds to the number of effective alleles of SNP ***i*** of sample ***j***, and can be 0, 1 or 2. Although PRSICE2 offers different additional options to calculate a PRS, these options (such as clumping, pruning, etc.) were omitted, and the PRS is calculated in this manner in order to be coherent with the reference analysis in [17]. Once the PRS for each study was obtained, in order to simplify the analyses further, the PRS values of the cohort were standardized to a Gaussian normal distribution by subtracting the mean of the PRS values of the entire cohort from each sample, and dividing by the standard deviation of the PRS values of the entire cohort.

We evaluated these standardized PRS values with the BC status of the samples to obtain a corresponding Receiver Operating Characteristic (ROC) Curve. We calculated the area under the ROC curve (AUC) to evaluate the performance of the different PRS models. Furthermore, we estimated the logistic regression-based odds ratio per 1 standard deviation of PRS (OR).

### 2.4. Hazard Ratio Calculation

Our next aim was to calculate the hazard ratio (HR) of the PRS related to the onset of BC, which will also be required for lifetime absolute risk estimation using iCare. For HR calculations using PRS values as the predictor, a right-censored and left-truncated Cox regression survival model was used. The Cox regression model is a useful approach to model time-to-event when the predictor is quantitative, as in our case [18]. In this model, the hazard score on predictor variable ***x***, which corresponds to the risk of time-to-event at time ***t***, can be obtained as
$$h(t,x)=\;\lambda_{0}(t)exp(b\;x)$$
where $\lambda_{0}$**(*t*)** corresponds to the baseline hazard function, which is common for all samples and can be obtained using the population incidence rate, ***x*** corresponds to a standardized PRS in our setting, ***b*** is the effect size, and exp(***b***) corresponds to the hazard ratio per 1 unit of standardized PRS (HR), and is independent from the baseline hazard function in the Cox regression model. In other words, for a standardized predictor ***x***, HR was obtained by calculating the ratio of the hazard score of samples, whose predictor value is 1 unit of standard deviation away from the mean to the hazard score of the sample whose predictor value is mean as
$$HR=\frac{h\left(t,1\right)}{h\left(t,0\right)}\;=\frac{\lambda_{0}\left(t\right)\;exp\left(b\;1\right)}{\;\lambda_{0}\left(t\right)\;exp\left(b\;0\right)}\;=exp\left(b\right).$$

Here, for calculating the HR of BC, the coxph.R function from the survival R package was used [30]. We chose the follow-up time as the time of diagnosis of BC for cases, and for controls we chose the time of the last health control. Once the hazard ratio was calculated, we also calculated the corresponding Herrel’s C-index (concordance index) that is a goodness-of-fit measure for models of hazard score calculated by the standardized PRS. 

### 2.5. Absolute Risk Estimation

Having obtained hazard ratio estimates, our next aim was to estimate the absolute BC onset risk of individuals by using the iCare tool based on the developed risk model [19]. According to this model, the individual risk is calculated in the presence of known risk factors (***x***), which in our case is the standardized PRS and the corresponding relative log HR parameters (***b***). Given these parameters, the model [19] can predict the next τ-year absolute risk for a current ***a*** year old individual as ($R\left(a,a+\tau \right)$**),**
$$R(a,a+\tau )=\int_{a}^{a+\tau }\lambda_{0}(t)\exp\left(b^{T}x\right)\exp\left(-\int_{a}^{t}\left[\lambda_{0}(u)\exp\left(b^{T}x\right)+m(u)\right]du\right)dt\;$$
where *m*(*t*) corresponds to the age-specific mortality rate function, and which was obtained from the Norwegian population age-specific mortality rate in 2016 from GHO data [31]. To build the model above and estimate the absolute risk of individuals, the required parameters are: log relative risks (***b***), which in our case was obtained from our log hazard ratio estimates, and the marginal age specific incidence rate $\lambda_{m}\left(t\right)$ to estimate $\lambda_{0}\left(t\right)$ as defined earlier, which was obtained from NORDCAN [32]. 

## 3. Results

### 3.1. Study Population and Genotypes

Initially we had 16,029 Norwegian individuals, of which 9201 had a cancer diagnosis. We excluded the samples diagnosed with other cancers and those without BC, samples with missing information, and males, resulting in the dataset containing 1053 cases and 7094 controls. We imputed genotypes for these samples with three different approaches and then calculated the PRS, HR and absolute lifetime risk.

Note that in this study we focused on examining the PRS as an independent risk factor for BC, and have not excluded the samples carrying pathogenic variants like *BRCA1* and *BRCA2* in order to be coherent with the reference study in [17]. The replication of the studies by excluding samples with these pathogenic variants is presented in Appendix A.

### 3.2. PRS Values

First, we calculated the PRS of these samples obtained for different PRS models [17], [5,13], and obtained the corresponding ROC curves by using standardized PRS values. As an illustrative example, the standardized PRS values obtained from PRSICE2 are presented in Figure 1A. As can be seen from this figure, there is a substantial shift in the mean of the distribution of cases compared to controls (0.381 and −0.039, respectively). Therefore, it is possible to use the standardized PRS values of the samples to predict the BC status of samples, as given in Figure 1B, with a ROC curve for data from the three different imputation pipelines. As can be seen in this figure, the ROC curves of all cases are far better than a random classifier (which corresponds to the purple straight line), while there is little difference between the imputation pipelines. Furthermore, as can be seen in Figure 1C, the four PRS models show a good correlation. In other words, a sample with a high PRS in one model is also highly likely to have a high PRS in another model. Therefore, in order to determine which PRS model and reference panel performs better, we calculated the AUC of these ROC curves and odds ratios (OR) with the three imputation pipelines in Table 1. In this table we have also included the results obtained from the Estonian Biobank (EstBB) and UK Biobank (UKB) data as provided earlier [17] to make a comparison across different populations.

The PRS model with 3820 SNPs gives the best performance compared to the others (Table 1). The corresponding AUC values in this model obtained for the samples imputed with 1000 GP, HRC, and Norgene reference panels were 0.621, 0620, and 0.625, respectively. These results are coherent with the one obtained from EstBB data (0.615) and UKB (0.632). Furthermore, we can observe that the Norwegian samples imputed from Norgene data perform better than the other reference panels (1000 GP and HRC). Although the improvement is small, one can observe a general trend of performance increase when a local reference panel is used (Table 1). A similar pattern can also be observed for the OR values.

### 3.3. Hazard Ratio Values

For modelling the time-to-event data (onset of BC) we used the standardized PRS values for all models as a predictor, with a right-censored and left-truncated Cox regression survival model. As given in Table 2, the hazard ratio (HR) per unit of standardized PRS (HR) was 1.494 (the best-performing case, which is using 3820 SNPs model, and imputing genome data using the Norgene reference panel), and the corresponding concordance index was also given as 0.607. Therefore, as observed for EstBB and UKB [17], we can state that the PRS is positively associated with the time-to-event of developing BC for the Norwegian population.

### 3.4. Absolute Risk Model

After the HR calculation, we estimated the lifetime individual risk of developing BC using the iCare tool [19]. In particular, we calculated the individuals’ lifetime risk of developing BC using the PRS of the samples as a risk factor, including age-specific incidence data and mortality rates. The age-specific incidence data for Norway were obtained from NORDCAN, and the age-specific mortality rates were obtained from the Global Health Observatory (GHO). Using these data with the calculated PRS values and HR, the absolute risk was determined for the samples whose PRS values are in the 1st, 25th, 50th, 75th, 90th and 99th percentiles. We used PRS values obtained from the four PRS models given above (all imputed with Norgene) and with a corresponding log hazard ratio based on the estimates. The results are presented in Figure 2. The calculated risks for the four different models show similar trends, while the risks in 3820 SNPs and 2803 SNPs are slightly higher. For a more detailed numeric evaluation, see Appendix A. Here we may focus on Figure 2A, corresponding to the 3820 SNPs model, which is nearly identical to that of 2803 SNPs (See Appendix A for the visual comparison). We can also observe a general trend in the other figures (Figure 2B–D) with slightly lower risks. In Figure 2A we observe that the cumulative risk of developing BC until the age of 80 is 21% for the samples in the 99th percentile and 14% in the 90th percentile. This risk reduces to 8% for the median and 3% for the 1st percentile. 

We also examined the risk of developing BC in the next 10 years using iCare, which can also be observed by taking derivatives of the cumulative risk curves in Figure 2. As expected, all models give similar risk values, as shown in Figure 3. If we focus on the age-based risk for 3820 SNPs model in Figure 3A, the absolute risk of developing BC in the next 10 years among 50-year-old women in the 1st percentile is 0.81%, while in the median percentile this risk is 2.23%, and in the 90th and 99th percentiles they are 3.5% and 5.2%, respectively. The same risk at the age of 35 for the samples in the 99th percentile is equal to the risk at the age of 40 for the 90th percentile, and these are equal to the risk at the age of 50 for the median percentile. 

Observed numbers of cases in various percentiles and deciles with respect to the PRS values are presented in Figure 4. Despite the number of samples being low for making strict conclusions, we can still observe that the number of cases increases in the higher percentiles/deciles. As can be seen in Figure 4A, the number of cases in the 99th percentile is higher than two times the number of cases for the median percentile, which is in line with the cumulative risks calculated using iCare. Furthermore, in Figure 4B, we have examined the number of cases per decile, and the number of cases in the highest decile is two times higher than the number of cases in the 5th decile. Therefore, samples in the highest deciles with respect to PRS can be considered as having more risk, both theoretically using the iCare tool and statistically using the actual information of the case samples.

## 4. Discussion

The main findings of this study are based on the application of existing Polygenic Risk Score (PRS) models for breast cancer (BC) in a Norwegian cohort. We show that the metrics of predicting BC using the PRS values of the Norwegian population, such as AUC, OR, and HR, are similar to findings in the samples from Estonia and UK Biobank [17]. We then calculated the lifetime PRS-based absolute risk of developing BC using iCare. The calculated risks are also in parallel with the risks obtained earlier [17]. Together, the current findings suggest a role for polygenic prediction models in BC screening in Norway.

Based on the AUC and OR values for the corresponding PRS models and reference panels, the Norwegian reference panel yields slightly better results compared to other reference panels applied to the Norwegian data. This may suggest that using local reference panels for imputation before calculating the PRS can be beneficial. On the other hand, if no local reference panel is available, the HRC and 1 kgp reference panels can be used to obtain the PRS values to detect BC at a comparable accuracy level. Regarding the PRS models used, the highest number of SNPs (3820) [5] gives the best performance among all models tested. The 2803 SNPs model, which has an available clinical implementation, demonstrates an almost identical performance to this model. 

Finally, we evaluated both individual and lifelong cumulative risk of developing BC in the next ten years using the iCare tool [19]. According to this risk model, samples whose PRS is in the highest decile have at least 2 times more risk than the samples whose PRS is around the mean. The risk of developing BC in the highest decile is evident much earlier in life than for the average patient. Therefore, for the samples with higher PRS values, starting screening at younger ages than the general recommendation for women (which is usually 50 years) could be beneficial. As suggested in [17], it may be practical to integrate a PRS into the existing BC screening for the Norwegian population. This can be undertaken by starting mammography screening at an earlier age, when a woman has a high PRS (hence high risks) compared to the population average.

Screening with mammography reduces BC mortality risk by 20–40% [33,34,35]. Current BC screening programs are based on age only, and mostly do not support regular screening of women below the age of 50. The relationship between the benefits and potential harms for screening in the age group under the age of 50 has been controversial, and screening is not recommended for all women under 50. The application of personalized risks is necessary to identify those women who could benefit from an earlier start to the screening.

Monogenic pathogenic variants and the PRS are the strongest independent risk factors for BC development [6,7]. An additional component of genetic susceptibility is also a family history without known MPV and PRS data. Genetic predisposition evaluations enable the separation of individual risk levels and open for precision, or stratified, screening and prevention activities. As there are other risk factors for BC besides genetic predisposition, the most practical way to use them for personalized screening depends on clinical application models.

With the help of the PRS, women can be divided into groups with different levels of risk based on the different recommendations for starting a mammography screening, or to whom other preventive measures can be given [20,36,37,38]. The PRS identifies women at a higher genetic risk who reach the threshold for population screening at a younger age, equivalent to risk for women aged 50 years who are eligible for population screening. The PRS also serves as an additional genetic risk evaluation tool for women who are negative for monogenic pathogenic variant findings in familial and breast cancer clinics. A study by Wolfson et al. has shown that population-wide programs for BC screening that seek to stratify women by their genetic risk should focus first on their PRS rather than the more highly penetrant but rarer variants, or their family history [6]. Considering these findings, it is expected that the PRS will be an important factor in BC screening in the near future.

Despite the PRS-based analysis in Norwegian data giving promising and similar results to other studies, we may not claim that the use of the PRS alone is sufficient to predict the existence or onset of BC. Considering the obtained performance metrics of the PRS such as AUC, OR, and HR, it is still likely that these can be improved by combining the PRS with other clinical factors. Therefore, one follow-up study would be to propose a harmonized prediction/stratification method that uses PRS scores and other possible factors that can be effective in BC prediction, such as the existence of *BRCA1* or *BRCA2* variants and/or some other clinical measures to obtain better prediction performances. 

One of the possible approaches is to use a Polygenic Hazard Score (PHS) model whose effectiveness has been proven in different age-related complex diseases, such as Alzheimer’s Disease (AD) [39], Parkinson’s disease [40], and Prostate Cancer [41,42], and which is also validated in Nordic samples for AD [43]. This approach integrates association with disease risk and age at onset into a common concept, under the hypothesis that genetic variation acts as a modulator of a lifetime risk. Applying this PHS model to our current dataset, we may obtain a predictive time-to-event model using the subset of predictive SNPs and other factors. This may attain an improvement in the prediction of the existence and onset of BC above what is presented here. 

Another possible follow-up study would be to examine the relation of the PRS, age of diagnosis and life expectancy after diagnosis. As can be observed in Appendix A, there is a distinguishable difference in risks, especially for *BRCA1*/*BRCA2* carriers, when we include *BRCA1*/*BRCA2* predisposing variants. Furthermore, it would also be also quite interesting and relevant to include the existence of other pathogenic variants [11]. Despite the fact that obtaining such comprehensive datasets can be challenging, the combination of these additional factors not only improves the performance of the prediction of BC, but also makes it possible to more precisely evaluate the lifetime risk or life expectancy, both with iCare and with the PHS. We are also planning to replicate the analysis in an independent dataset. Such a follow-up study is also relevant, since the samples in this study are collected from the families with known cancer histories. Therefore, the effect of such a biased data collection could also be evaluated when similar analyses are conducted using an independent dataset. 

To summarize, our findings suggest that the PRS-based risk evaluation can be a useful and complementary method in the screening for BC, especially for younger age groups, to identity relatively few but nevertheless high-risk individuals in Norway. Our findings may open a new perspective on a modified screening strategy for BC in Norway by determining the start time of the actual screening process based on the PRS scores, as suggested [17]. It is also possible to enhance the presented detection and stratification approaches in different aspects to obtain a more precise screening strategy.

## Figures and Tables

**Figure 1 cancers-15-04124-f001:**
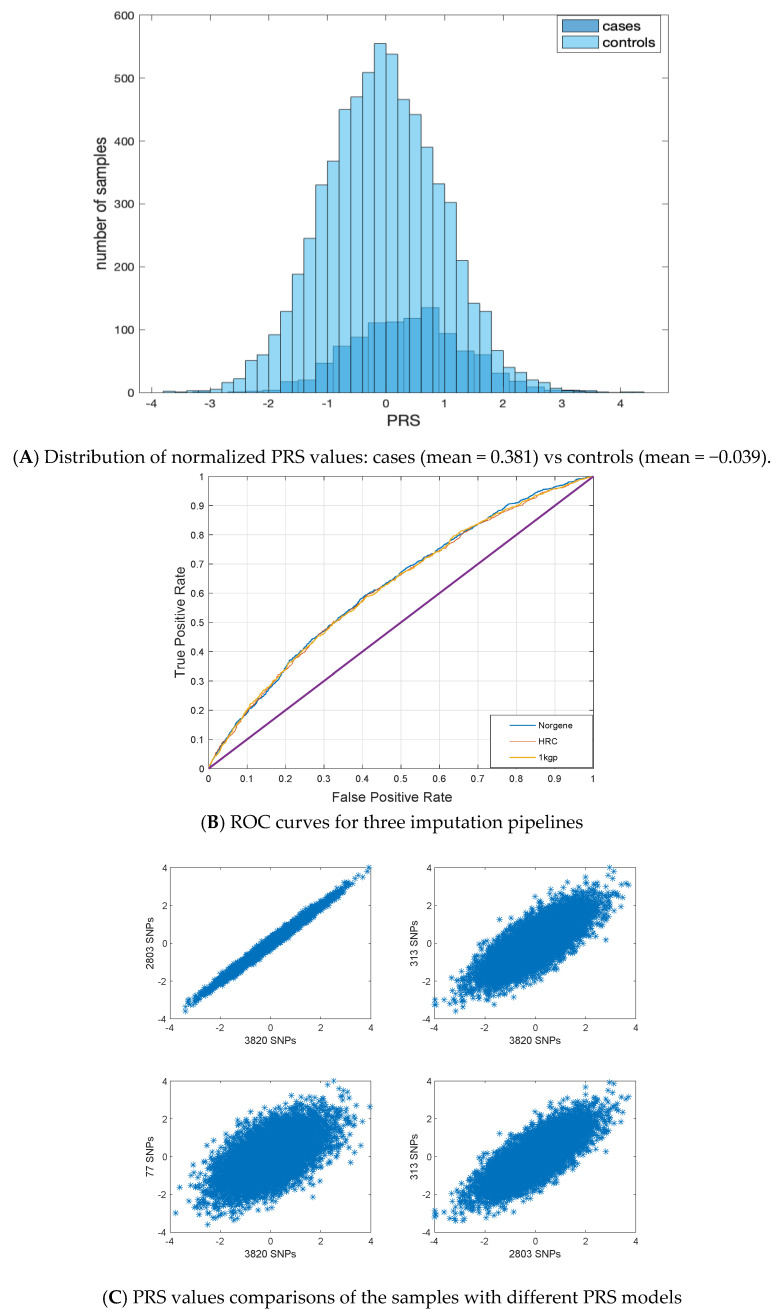
Illustration of (**A**) the distributions of polygenic risk score (PRS) values of cases and controls, (**B**) receiver operating characteristic (ROC) curves obtained using different imputation pipelines (Table 1) for the 3820 SNPs model in the Norwegian sample. Norgene, HRC, and 1 kgp corresponds to the reference panels used to impute data. (**C**) Correlation plots of the PRS values of the samples across the studied PRS models.

**Figure 2 cancers-15-04124-f002:**
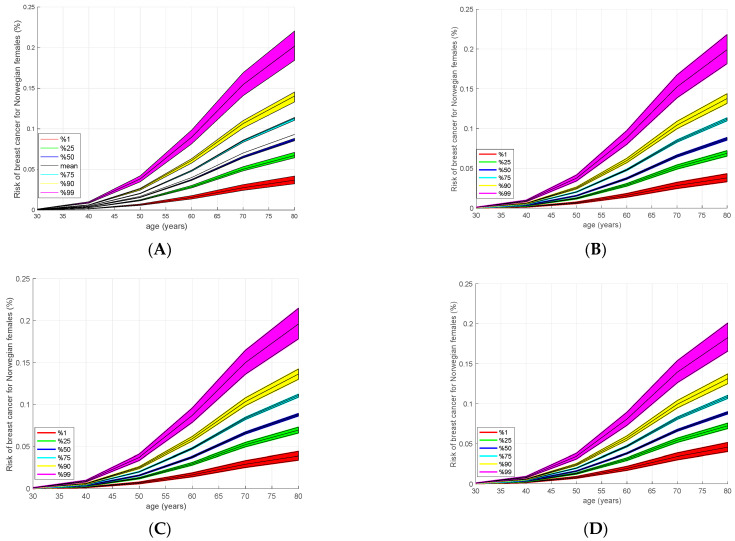
Calculated cumulative BC risks obtained from iCare using four different PRS models. The disease incidence rate is obtained from Nordcan [32]. Calculated absolute risks are almost the same for the 3820 SNPs and the AnteBC (2803 SNPs), while there is a slight decrease in the risk when other models are used. On the other hand, all of the models give similar trends with respect to risk groups and age. Typically, the life-long cumulative risk of the 99th percentile is almost 2.5 times the risk of the median samples, and 6 times the risk of the 1st percentile. (**A**) 3820 SNPS. (**B**) AnteBC (2803 SNPs). (**C**) 313 SNPs. (**D**) 77 SNPs.

**Figure 3 cancers-15-04124-f003:**
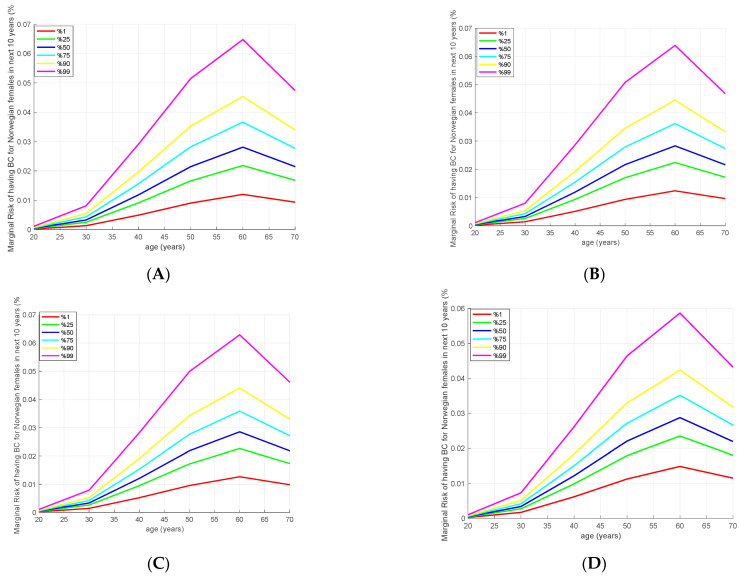
The risk of developing breast cancer (BC) in the next 10 years for four different polygenic risk score (PRS) models. (**A**) 3820 SNPS. (**B**) AnteBC (2803 SNPs). (**C**) 313 SNPs. (**D**) 77 SNPs.

**Figure 4 cancers-15-04124-f004:**
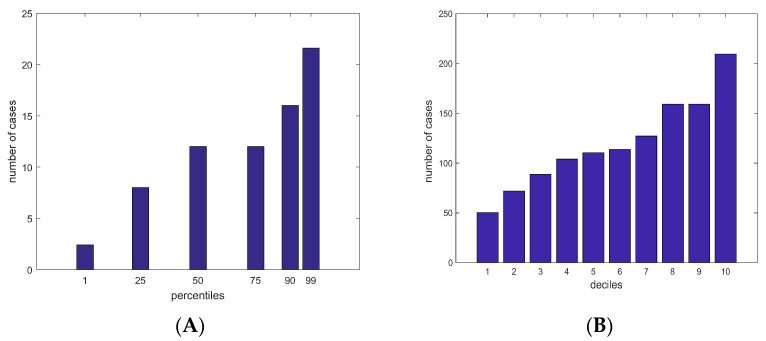
The observed number of cases in our dataset per percentiles and deciles with respect to PRS. (**A**) Number of cases per percentiles. (**B**) Number of cases per deciles.

**Table 1 cancers-15-04124-t001:** The polygenic risk score (PRS) performance of Norwegian data for breast cancer (BC) with different imputation pipelines (shaded columns), and their comparison with the Estonian Biobank (EstBB) and UK Biobank (UKB) data. Note that in these calculations we have excluded samples that have any other cancers and those without BC, resulting in 1053 samples for cases and 7094 for controls. OR: odds ratio, AUC: Area under curve.

Metrics	SNPSET	Antegenes Pipeline (1 kgp)	MoBa (HRC)	Norgene (Norwegian Reference Panel)	EstBB (Estonian Reference Panel)	UKB
**AUC (number of SNPs used)**	**77 SNPs**	**0.582 (66)**	**0.584 (66)**	**0.600 (66)**	**0.591 (73)**	**0.607 (73)**
	**313 SNPs**	**0.605 (232)**	**0.605 (210)**	**0.617 (228)**	**0.604 (257)**	**0.625 (257)**
	**2803 SNPs**	**0.616 (2511)**	**0.617 (2379)**	**0.618(2474)**	**0.615(2803)**	**0.632 (2803)**
	**3820 SNPs**	**0.621 (2706)**	**0.620 (2482)**	**0.625 (2698)**	**0.611(3081)**	**0.632 (2803)**
**OR (SE)**	**77 SNPs**	**1.460 (0.034)**	**1.465 (0.034)**	**1.503 (0.033)**	**1.369 (0.061)**	**1.485 (0.012)**
	**313 SNPs**	**1.465 (0.034)**	**1.475 (0.034)**	**1.534 (0.034)**	**1.426 (0.060)**	**1.556 (0.012)**
	**2803 SNPs**	**1.526 (0.034)**	**1.538 (0.034)**	**1.534 (0.034)**	**1.479 (0.061)**	**1.616 (0.012)**
	**3820 SNPs**	**1.546 (0.034)**	**1.528 (0.034)**	**1.567 (0.034)**	**1.474 (0.060)**	**1.617 (0.012)**

**Table 2 cancers-15-04124-t002:** PRS-based Cox regression survival analysis performance of Norwegian data for BC with different imputation pipelines (shaded columns) and their comparison with EstBB and UKB. Note that in these calculations we have excluded controls with any other cancer/cancers and those without BC, resulting in 1053 samples for cases and 7094 for controls.

SNPSET	Metrics	Antegenes Pipeline (1 kgp)	MoBa (HRC)	Norgene (Norwegian Reference Panel)	EstBB (Estonian Reference Panel)	UKB
**77 SNPs**	**HR (%95 confidence interval)**	1.273 (1.211–1.339)	1.276 (1.214–1.342)	1.373 (1.307–1.443)	1.580 (1.428–1.747)	1.440 (1.406–1.476)
	**c-index**	0.563 (se = 0.008)	0.563 (se = 0.008)	0.581 (se = 0.008)	0.638 (se = 0.015)	0.600 (se = 0.004)
**313 SNPs**	**HR (%95 confidence interval)**	1.373 (1.305–1.444)	1.355 (1.288–1.425)	1.439 (1.368–1.513)	1.615 (1.457–1.789)	1.551 (1.515–1.588)
	**c-index**	0.58 (se = 0.010)	0.578 (se = 0.008)	0.593 (se = 0.008)	0.642 (se = 0.015)	0.622 (se = 0.004)
**2803 SNPs**	**HR (%95 confidence interval)**	1.421 (1.351–1.494)	1.442 (1.370–1.517)	1.455 (1.384–1.531)	1.660 (1.500–1.837)	1.562 (1.526–1.588)
	**c-index**	0.593 (se = 0.008)	0.596 (se = 0.008)	0.598 (se = 0.008)	0.656 (se = 0.015)	0.625 (se = 0.003)
**3820 SNPs**	**HR (%95 confidence interval)**	1.462 (1.375–1.554)	1.458 (1.371–1.55)	1.494 (1.406–1.588)	1.654 (1.494–1.830)	1.562 (1.526–1.600)
	**c-index**	0.602 (se = 0.010)	0.601 (se = 0.010)	0.607 (se = 0.010)	0.654 (se = 0.015)	0.625 (se = 0.003)

## Data Availability

In accordance with Norwegian legislation and the ethical approval of the study by the Regional Committee for Medical and Health Research Ethics, South-Eastern Norway, the raw high-throughput DNA sequencing and genotyping data generated in this study are considered patient identifiable and subject to secure storage regulations in accordance with the national Personal Data Regulations, chapter 2. Data will be made available upon reasonable request, subject to compliance with the consent requirements restricted to cancer, and this will require the formalization of a data transfer agreement.

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
