# Peer review of "A Breast Cancer Polygenic Risk Score Is Feasible for Risk Stratification in the Norwegian Population"

_cancers, 2023, doi:10.3390/cancers15164124_

Round 1

Reviewer 1 Report

The authors present the evaluation of the performance of four polygenic risk scores for breast cancer in a sample set from Norway. Their findings are in line with other previous studies suggesting that the PRS has a similar performance as in other populations and potentially can be used to guide clinical practise in Norway too. They find that the best performing PRS which includes 3820 SNPs yielded an AUC = 0.625 and OR per standard deviation OR = 1.567.

The manuscript is of interest but needs some editing to give more clarity to the audience.

Are these data highly selected? The authors state that of 9201 cancer patients 3223 had germline pathogenic variants which is very high percentage – were the individuals carrying pathogenic variants removed?

The authors comment in the text about  family history and pathogenic variant status but have not combined these with the PRS. It will be interesting to see if there is any differentiation of the PRS by family history or pathogenic variant status.

Methodology:

-          Line 152: which mean and standard deviation?

-          Line 255: do you mean Supplementary tables?

-          Line 256 – the results might be identical but the model cannot be right? 3820 SNPs vs 2803?

Other minor comments:

-          Affiliations don’t follow correct numbering

-          The format of text and some of the figures is misplaced

-          The fonts are not the same throughout the text

-          The decimal numbers used should be the same throughout the manuscript.

Some rewording/rewriting needs to be performed so there is more clarity in the manuscript:

-          I am not sure you can develop “risk monitoring methods” with PRS, maybe reword to “risk classification or risk categorization..”

-          Line 18 “BC in Norwegian population and can be BC screening programme.” Something is missing

-          The performance of a PRS is yet uncertain  - I think uncertain is not the right word, maybe unknown

-          Line 33: “in this Norwegian sample” maybe in this study from Norway or something

-          Line 138:  “were tested for earlier” do you mean were tested in previous studies?

-          Line 275 – as shown in Figure 3?

-          Line 317/8 – something is missing

-          Line 390 “may open” – something is missing

-          Mutations should be replaced with pathogenic variants

Author Response

We would like to thank the reviewer for such valuable critics which will improve the quality of our study considerably. The responses can be found in the file attached as "reviewer1_response.docx" and also in Cover Letter sent to the editor.

Reviewer 2 Report

Major comments:  How are these PRS models compare to each other on the risk stratification performance? In other words, how conisistent are individuals identified by one PRS approach as high risk also being identified by another one?    The author mentioned Norwegian population, it would be interesting and at least to show the GWAS restuls in this population first before constructing PRSs. Also, it would be interesting to the minor allele frequency of the SNPs being included in Norwegian population compared to other populations, and how they are consistent or different.    It’s not clear to me the "longitudinal risk” and "age-dependent”. Does this refer to predicting the incident cases of breast cancer vs. the prevalent breast cancer? More clarification is needed.    It seems the authors didnt include other covariates in predicting risk of breast cancer, which is highly problematic and decreases the credibility of PRS in real-world setting. Also, it would be important to have stratified analyses by family history of breast cancer, BRCA1, 2 status, etc.    minor comments:  different fonts being used in the abstract    For the simple summary part, more concrete results and conclusions are necessary.  Consistent rounding should be applied throughout the manuscript

Author Response

We would like to thank the reviewer for such valuable critics which will improve the quality of our study considerably. The responses can be found in the file attached as "reviewer2_response.docx" and also in Cover Letter sent to the editor.

Round 2

Reviewer 2 Report

Thanks for addressing my concerns and revising the manuscript.